# A Bipartite Geminivirus with a Highly Divergent Genomic Organization Identified in Olive Trees May Represent a Novel Evolutionary Direction in the Family *Geminiviridae*

**DOI:** 10.3390/v13102035

**Published:** 2021-10-09

**Authors:** Patrick Materatski, Susan Jones, Mariana Patanita, Maria Doroteia Campos, António Bento Dias, Maria do Rosário Félix, Carla M. R. Varanda

**Affiliations:** 1MED—Mediterranean Institute for Agriculture, Environment and Development, Instituto de Investigação e Formação Avançada, Universidade de Évora, Pólo da Mitra, Ap. 94, 7006-554 Évora, Portugal; mpatanita@uevora.pt (M.P.); mdcc@uevora.pt (M.D.C.); carlavaranda@uevora.pt (C.M.R.V.); 2Information and Computational Science Department, The James Hutton Institute, Dundee DD2 5DA, UK; sue.jones@hutton.ac.uk; 3MED—Mediterranean Institute for Agriculture, Environment and Development & Departamento de Engenharia Rural, Escola de Ciências e Tecnologia, Universidade de Évora, Pólo da Mitra, Ap. 94, 7006-554 Évora, Portugal; adias@uevora.pt; 4MED—Mediterranean Institute for Agriculture, Environment and Development & Departamento de Fitotecnia, Escola de Ciências e Tecnologia, Universidade de Évora, Pólo da Mitra, Ap. 94, 7006-554 Évora, Portugal; mrff@uevora.pt

**Keywords:** high-throughput sequencing, *Olea europaea* L., *Geminiviridae*, recombination, evolution

## Abstract

Olea europaea Geminivirus (OEGV) was recently identified in olive in Italy through HTS. In this work, we used HTS to show the presence of an OEGV isolate in Portuguese olive trees and suggest the evolution direction of OEGV. The bipartite genome (DNA-A and DNA-B) of the OEGV-PT is similar to Old World begomoviruses in length, but it lacks a pre-coat protein (AV2), which is a typical feature of New World begomoviruses (NW). DNA-A genome organization is closer to NW, containing four ORFs; three in complementary-sense AC1/Rep, AC2/TrAP, AC3/REn and one in virion-sense AV1/CP, but no AC4, typical of begomoviruses. DNA-B comprises two ORFs; MP in virion sense with higher similarity to the tyrosine phosphorylation site of NW, but in opposite sense to begomoviruses; BC1, with no known conserved domains in the complementary sense and no NSP typical of bipartite begomoviruses. Our results show that OEGV presents the longest common region among the begomoviruses, and the TATA box and four replication-associated iterons in a completely new arrangement. We propose two new putative conserved regions for the geminiviruses CP. Lastly, we highlight unique features that may represent a new evolutionary direction for geminiviruses and suggest that OEGV-PT evolution may have occurred from an ancient OW monopartite *Begomovirus* that lost V2 and C4, gaining functions on cell-to-cell movement by acquiring a DNA-B component.

## 1. Introduction

Olive (*Olea europaea* L.) is one of the most cultivated fruit crops around the world and has significant environmental, social and landscape impact in many countries such as Portugal. Olive trees are known to be infected by 17 different viruses [1,2,3,4,5,6,7,8,9]. A common feature of olive viruses is the difficulty of associating viral presence to symptoms. Some authors have associated viral symptoms with chlorosis, defoliation, bumpy fruits, reduction of yield and oil quality [10,11], however, most viruses, have been recovered from trees without apparent symptoms [7,10,12]. Nevertheless, despite the difficulty in associating viruses to symptoms, many olive viruses are easily transmitted to other hosts and olive propagative material must be free of viruses for certification and commercialization. For these reasons, studies of the olive virome in different countries is of great economic importance.

OEGV (Olea europaea geminivirus) is the most recently identified virus in olive [8]. It was identified in olive trees in Italy using high-throughput sequencing (HTS) and belongs to the family *Geminiviridae* that is the largest family of plant viruses alongside the family *Potyviridae*. Members from the family *Geminiviridae* are present in tropical and subtropical regions and constitute over 520 distinct species [13,14]. Geminiviruses have either one or two circular single-stranded DNA (ssDNA), with approximately 2.7–5.2 kb, encapsidated into geminate icosahedral virions [15]. Very recently, this family increased from nine recognized genera: *Becurtovirus*, *Begomovirus*, *Capulavirus*, *Curtovirus*, *Eragrovirus*, *Grablovirus*, *Mastrevirus*, *Topocuvirus*, *Turncurtovirus* to 14 genera [16], with *Citlodavirus*, *Maldovirus*, *Mulcrilevirus*, *Opunvirus* and *Topilevirus* being added. Besides these genera, there are still a number of divergent geminivirus species that have not yet been assigned to any genus [8,17]. HTS technologies have contributed to the recent extension of the *Geminiviridae* family and increased our understanding of the evolutionary history of this family.

Genera *Begomovirus* and *Mastrevirus* correspond to 94% of the *Geminiviridae* virus species, although *Begomovirus* is by far the genus with the highest number of recognized species, over 445 [16]. This remarkable success, in the higher number of species, can be attributed to many features including high prevalence of the virus vector *Bemisia tabaci* in monocultures worldwide and infection of non-cultivated plants. In addition, these viruses have the capacity to rapidly evolve via pseudo- and recombination, mutation and acquisition of new DNA components and satellites [18,19,20]. Begomoviruses are classified into two main lineages, Old World (OW: Eurasia, Africa and Oceania) and New World (NW: the Americas). Most NW *Begomovirus* are bipartite, comprising DNA-A and DNA-B, with a few monopartite reported exceptions [21,22]. The DNA-A of bipartite NW begomoviruses has five open reading frames (ORFs), one in the virion-sense encoding the coat protein (AV1 or CP) and four in the complementary sense encoding the replication initiation protein (AC1 or Rep), the transcriptional activator (AC2 or TrAP), the replication enhancer (AC3 or REn) and the AC4 protein (or C4) [15,23]. The DNA-B of bipartite NW begomoviruses comprises two ORFs; one in the virion-sense encoding the nuclear shuttle protein (BV1 or NSP); one in the complementary sense encoding the movement protein (BC1 or MP) [19,24]. In contrast with the NW begomoviruses, the majority of OW begomoviruses are monopartite with their DNA comprising 6 ORFs, presenting an additional ORF V2 or AV2, also referred as pre-coat [25], and that is not present in NW begomoviruses. It was believed that NW *Begomovirus* appeared more recently than OW *Begomovirus*, after continental separation in the Cenozoic era [26], however, some studies have provided evidence of the NW Begomoviruses’ presence in the Old World, prior to continental separation [27].

DNA-A and DNA-B of begomoviruses are very different in terms of nucleotide sequence, except in part of the intergenic region (IR) segment (~180 nt) known as “common region” (CR). Inside the CR, they present an origin of replication (Ori), that includes a stem–loop structure (~30 nt) containing a nonanucleotide sequence (5′-TAATATTAC- 3′) typical of most *Geminiviridae* and conserved iterative sequences (iterons). These bidirectional iterative sequences (5–8 nt in length) are required for specific recognition and binding by Rep during replication [28,29,30]. The T–A site within the conserved nonanucleotide sequence is necessary during replication for cleaving and joining viral DNA. Together, the conserved nonanucleotide and the iterons are functional targets for Rep, that recognizes and binds specifically to the iterons and subsequently introduces a nick into the nonanucleotide sequences to initiate the DNA replication by a rolling-circle (RCR) mechanism [30,31].

The remaining *Geminiviridae* virus species cluster into 13 genera that have exclusively monopartite genomes and differ in genome arrangement. They produce between 4 and 8 multifunctional proteins, encoded by bidirectional and partially overlapping ORFs. Functions of these proteins have mostly been determined by positional analogy to other studied proteins within the *Geminiviridae* and are putatively involved in a variety of functions such as encapsidation, viral movement, vector transmission, viral replication, plant cell interference, regulation of viral gene expression and suppression of host antiviral responses [15,23]. Members within the *Geminiviridae* present many mixed features among them, reinforcing that recombination is a major evolution driver within this family. For example, *Becurtovirus* CP is *Curtovirus*-like whereas the Rep is most related to that of *Mastrevirus*. *Eragrovirus* C2 is a positional analog of *Topocuvirus*, *Curtovirus* and *Begomovirus* TrAP genes, Rep is most related to *Begomovirus*, *Curtovirus* and *Topocuvirus* and CP is most similar to *Mastrevirus*. *Topocuvirus* genome is similar to monopartite *Begomovirus* and are suggested to have diverged after a recombination event that altered vector specificity [32], originating a CP more closely related to *Curtoviruses*. *Turncurtovirus* have a similar genome arrangement in the complementary sense to that of *Curtoviruses*, however, with low sequence identity, except for C4.

OEGV, first identified in olive in Italy, was classified as a putative member of a new genus within the *Geminiviridae* family [8]. In our study, we used RNA-sequencing to identify OEGV in olive in Portugal (denoted in this paper as OEGV-PT) and we provided new evidence that OEGV-PT has genomic characteristics from both OW and NW *Begomovirus*. We also show that OEGV-PT presents unique features distinct from other geminiviruses and suggest that OEGV-PT evolution may have occurred from an ancient OW monopartite *Begomovirus* that lost V2 and C4, gaining functions on cell to cell movement by acquiring a DNA-B component.

## 2. Materials and Methods

### 2.1. Sampling Protocol

Sampling was carried out at the end of the autumn and at the beginning of the winter in 2019. Four olive oil producing sites were sampled; three in the Alentejo region, south of Portugal (sites A, B and C) and one in the Ribatejo region, center of Portugal (site D); all sites are influenced by Mediterranean climate. Sampled olive groves occupy an area of 320,000 m^2^ in Monforte (site A) (39°4′3.99″ N, 7°28′13″ W), 450,000 m^2^ in Évora (site B) (38°30′05.54″ N, 7°45′19.79″ W), 150.000 m^2^ in Safara (Site C) (39°4′3.99″ N, 7°16′13″ W) and 180,000 m^2^ in Santarém (site D) (39°17′04.81″ N, 8°41′24.99″ W). The olive trees were 10–30 years old, produced under intensive regime (planted with a spacing of 7 × 5 m), or super intensive regime (planted with a spacing of 4 × 1.35 m) and belonged to four different cultivars (Galega vulgar, Cobrançosa, Picual and Arbequina). At each site, 10 olive trees of each cultivar (10 trees × 4 cultivars) were randomly sampled. Each sample consisted of 10 3-year stems cut from each plant around the whole tree at a height of 1.5 m above ground. All sites received programmed applications of phytosanitary treatments such as fungicides; benzimidazole or MBC, demethylation inhibitor (DMI), strobilurin, copper hydroxide and copper oxychloride, and insecticides; lambda-cyhalothrin, dimethoate and deltamethrin. A total of 160 trees were sampled (4 sites × 4 cultivars × 10 trees). Samples were transported to the laboratory in a refrigerated basket, stored at 4 °C and processed within the next 24 h. Cortical scrapings of the 3-year stems of each sampled tree were ground in liquid nitrogen, mixed in equal amounts, combined per cultivar at each site (4 cultivars × 4 sites) and stored at −80 °C.

### 2.2. RNA-Extraction and Sequencing

Total RNA of each of the 16 pooled samples was extracted and purified using Qiagen RNeasy Plant Mini kit following the manufacturer’s instructions. Purified RNA was eluted in RNAse-free water and stored at −80 °C prior to subsequent sequencing at the Fera Science Ltd. (York, UK). Purified RNAs were subtracted from the ribosomal RNA using a Ribo-Zero rRNA removal kit for plants (Illumina, San Diego, CA, USA). Subsequently, cDNA libraries were prepared using a ScriptSeq library preparation kit (Illumina) and paired-end sequencing was conducted using an Illumina MiSeq (2 × 300 nt version 3) machine. The controls included an uninfected tobacco leaf spiked with External RNA Control Consortium (ERCC) positive control artificial RNA.

### 2.3. Bioinformatics Analysis of RNA-Sequence Data

Raw reads were trimmed using Trimmomatic [33] with the following parameters: LEADING:20 TRAILING:20 SLIDINGWINDOW:4:20 and a minimum read length of 30. To remove host reads, trimmed reads were mapped to the concatenated genome sequences of *Olea europaea* var. *sylvestris* (NC_036237.1) and Olive chloroplast (FN996972.1). Mapping was conducted using Bowtie2 (v2.3.4.3)^3^ [34] (score-min value “L, 0, −0.2”). The unmapped reads, designated as non-host reads, were assembled into contigs using Trinity software (v2.8.4)^4^ [35]. Contigs >200 bp in length were then mapped against a database of 1,461,177 proteins, derived from 39,163 viruses with complete genomes in Genbank (v237) [36]. The mapping was conducted using DIAMOND software6 (with parameters --max-target-seqs 5 --e-values <10-6 --subject-cover 25 --sensitive). The predicted virus sequences were then filtered at either (a) 50% protein sequence identity and 25% target protein length overlap or (b) 30% sequence identity, a 25% target overlap length at the protein level and a contig length > 1000 bp.

### 2.4. DNA Extraction, PCR Validation, and Full-Length Amplification of Viral Genomic DNA Molecules

After the identification of a DNA virus (geminivirus) sequences in the RNA-sequence results, the total DNA of each olive sample was extracted and purified using a DNeasy Plant Mini kit (Qiagen, Hilden, Germany) following the manufacturer’s protocol, for the subsequently polymerase chain reaction (PCR) validation in the samples. DNA concentration was determined by using a Quawell Q9000 micro spectrophotometer (Quawell Technology, Beijing, China). First, the PCR validation was done based on the two divergent contig sequences (2727 and 1286 bp) mapped by HTS, using two specific pairs of primers, in addition, several PCR specific primers were designed based on the two different contigs (2727 and 1286 bp) to recover full-length DNAs (Appendix A). The strategy was based on the inverse PCR assays to bi-directionally (due to the circular DNA) amplify the full-length DNAs. PCR reactions were performed in a total volume of 50 μL, containing 30–80 ng of genomic DNA, 10 mM Tris-HCl (pH 8.6), 50 mM KCl, 1.5 mM MgCl_2_, 0.2 mM dNTPs (Fermentas, Thermo Scientific, Waltham, MA, USA), 0.2 μM of each primer and 2.5 U of DreamTaq DNA polymerase (Fermentas, Thermo Scientific, Waltham, MA, USA). Amplification reactions were carried out in a Thermal Cycler (BioRad, Hercules, CA, USA) with an initial temperature of 95 °C for 2 min, followed by 40 cycles of 95 °C for 30 s; 55 °C for 1 min and 72 °C for 2 min, as well as a final extension at 72 °C for 10 min. Amplified products were first analyzed by electrophoresis in 1% agarose gel and then purified using DNA Clean & Concentrator (Zymo Research, Irvine, CA, USA) according to the instructions of the manufacturer, and finally sequenced by Macrogen (Madrid, Spain) in both directions. BLAST, at the National Center for Biotechnology Information (NCBI) was used as a basic local alignment search tool to search for homologous sequences.

### 2.5. Sequence and Phylogenetic Analyses

MEGA software version 10.1.8 [37] with CLUSTAL W was used for the analysis of nucleotide and amino acid sequences of OEGV-PT DNA-A and DNA-B. Sequences from 39 members, most representative of the family *Geminiviridae*, including some viruses that had not yet been classified into genera, were retrieved from the GenBank database (release 242.0). Virus genera, species and abbreviations, and GenBank accession numbers used are summarized in Appendix A. The identification of predicted proteins encoded by each genomic DNA molecule (DNA-A and DNA-B) in OEGV, was achieved using Open Reading Frame (ORF) Finder (RRID:SCR_016643) (https://www.ncbi.nlm.nih.gov/orffinder; accessed on 6 September 2021) and the SmartBLAST algorithm at NCBI. The ProtParam program was used for determination of the theoretical isoelectric point and protein molecular mass [38]. For multiple sequence alignments (nucleotide and amino acid) the Clustal Omega alignment program (European bioinformatics Institute, EMBL-EBI) was used. Phylogenetic trees were constructed using the maximum-likelihood (ML) method based on the Tamura–Nei model, that showed the lowest Bayesian information criterion (BIC) score and was the best-fit nucleotide/amino acid substitution model for these data. To establish relationships between sequences according to their genetic distances the neighbor joining (NJ) model was used. The significance of the interior branches was determined through bootstrap analyses with 1000 replicates. Pairwise sequence identities of the full genome, CP and Rep of OEGV-PT and representatives of different geminiviruses lineages were determined using Sequence Demarcation Tool (SDT) v1.2 [39].

### 2.6. Recombination Analysis

To detect evidence of recombination in OEGV-PT, we analyzed OEGV-PT DNA-A sequence together with the previously described 39 additional geminivirus genome sequences (Appendix A) and DNA-B sequence with 12 other begomoviruses. Sequences were aligned with MEGA 10.1.8 [37] and recombination analysis was performed using the Recombination Detection Program (RDP4.101 Software) [40] with default settings using the detection methods RDP, GENECONV, BOOTSCAN, MaxCHI, CHIMERA, SiSCAN and 3SEQ. Only recombination events detected by three or more methods with *p*-values < 0.05 were accepted.

### 2.7. Prediction of Insect Vector through CP Analysis

CP amino acid sequences were used to search for specific patterns within subgroups of viruses that shared the same type of insect vector for prediction of the putative insect vector of OEGV-PT. Protein specificity determining positions (SDPs) were determined using Speer-server [41] and SCI-PHY for automated subgrouping [42], setting the relative entropy term and physico-chemical (PC) property distance term weights to one.

## 3. Results

### 3.1. RNA-Sequence Analysis

The total number of trimmed sequences reads recovered from the 16 samples ranged from 882K to 2.1 million. *Olea europaea* var. *sylvestris* (NC_036237.1) and Olive chloroplast (FN996972.1) (host reads) constituted between 42% and 55% of the trimmed reads in each olive sample. Assembling the non-host reads into contigs, gave between 743 and 21.9K contigs per sample. The number of contigs >2500 bp length was small and ranged between 0 and 67 (Table 1). Contigs >200 bp in length were mapped against viral genomes in Genbank and six contigs mapped to sequences from the *Geminiviridae* family (Table 1). From these, two were in trees from the cultivar Galega vulgar at site C, and comprised 2727 and 2475 bp, respectively, and three were also from the cultivar Galega vulgar but at sites B (two contigs) and D (one contig) and comprised 808, 874 and 323 bp, respectively. These five contigs showed 100% identity. One remaining contig (1286 bp), also in the cultivar Galega vulgar at site C, shared only 63.6% sequence similarity to the other 5 contigs. A BlastP search against the Genbank virus genomes with the 2727 and 1286 bp length contigs, showed 100% identity in DNA-A (accession number MW316657; 2775 bp in length) and 99.96% identity in DNA-B (MW316658; 2763 bp in length), which both correspond to a geminivirus found in olive in Italy and named Olea europaea Geminivirus (OEGV) [8].

### 3.2. Validation of HTS Results and OEGV-PT Presence in Olive

RNA-sequence derived results were validated through the amplification of the two different contigs (2727 and 1286 bp) using a pair of primers specifically designed for each contig. The recovery approach of the full-length genome of the OEGV-PT DNA (with a circular DNA) was done using the set of overlapping primers specifically designed for each of the contigs sequences and followed by inversed PCR primers assays. The full nucleotide (nt) sequencing of both contigs (2727 and 1286 bp) revealed two circular DNAs; of 2775 and 2763 bp, respectively, presenting the same length of OEGV genomes described by Chiumenti et al. [8]. In addition, the validation of olive samples showed that a total of 27 (16.9%) from the 160 sampled trees were positive for OEGV-PT DNA (DNA-A and DNA-B). None of the trees sampled had typical viral symptoms. At site B a total of 8 (20%) olive trees were positive; at site C there were 10 positive (25%) and at site D there were 9 positive (22.5%). All trees at site A were negative. An intriguing result was that all positive trees belonged to the cultivar Galega vulgar, suggesting a certain specificity of OEGV-PT for this cultivar, however, more robust sampling, in terms of number of trees, fields, or regions is needed to validate this observation.

### 3.3. Genome Organization of OEGV-PT DNA-A and Its DNA-B Cognate Molecule

The nucleotide (nt) similarity, size and genome organization of OEGV-PT DNA-A is identical to OEGV isolate from Italy [8]. Interestingly, and also as verified by Chiumenti et al. [8] no C4, typical of the begomoviruses, was found. The full nucleotide (nt) sequence of OEGV-PT DNA-A (2775 bp) presents a length typical of OW begomoviruses and an organization similar to NW begomoviruses. The DNA-B of OEGV-PT presents the same length (2763 bp) and genome organization of OEGV isolate from Italy [8] but differs in a single nucleotide within the BV1/MP (Appendix A).

Based on the alignment of DNA-A and DNA-B genomic sequences (Figure 1A), the longest identical sequence segment flanking the invariant nonanucleotide sequence was considered as the common region (CR). OEGV-PT DNA-A and DNA-B share a CR of approximately 403 nt (2493–121 nt), which is 99% identical, with only three polymorphisms, evidence that these are cognate components of a bipartite genome. The CR region contains a structure that has the characteristics of an inverted repeat capable of forming a stem-loop structure (Figure 1) and also harbors a highly conserved apex with invariant nonanucleotide 5′-TAATATT↓AC-3′ (the Rep-nicking site), and an GC rich region close to the stem-loop structure was identified. In addition, there was a TATA box and four replication-associated iterative sequences “iterons”; two sites with a TGGGGA consensus upstream the TATA box and two inverted repeat sequences TCCCCA, one downstream and the other upstream from the TATA box (Figure 1A,B).

### 3.4. Phylogenetic Relationships and Pairwise Identities with Other Geminiviruses

OEGV-PT DNA-A was shown to be 98–100% identical to 11 full DNA-A sequences recently deposited in Genbank (Accession numbers MW316657 and MW560446 to MW560455) [43], corresponding to a geminivirus found in olive and to which the name Olea europaea Geminivirus (OEGV) was proposed [8].

OEGV-PT DNA-A pairwise and phylogenetic analysis of AC1 and AV1 predicted proteins confirms divergent results in terms of geminiviral origin as suggested by Chiumenti et al. [8] (Figure 2A). We included the analysis of AC2 and AC3 proteins, both showing to be more related to *Begomovirus*, as verified for AC1 (Appendix A). We also included, in this analysis, a new group of NW begomoviruses, belonging to the SLCV clade with which OEGV-PT DNA-A showed the highest similarity (62.1%). OEGV-PT full DNA-B sequence is 99.96% similar to OEGV full DNA-B from Italy (Accession number MW316658), with a single difference at nt 981, C in the Italian and T in the PT isolate.

OEGV and OEGV-PT DNA-B identity to the other begomoviruses ranges from 57.1% to 60.5%. The translation product of ORF BC1 (534 nt; 177 aa; 24.1 kDa) showed no resemblance in the blastP analysis. The translation product of ORF BV1 (891 nt; 296 aa; 34.5 kDa) is related to *Begomovirus* BC1.

The translation product of ORF AC1 (1.095 nt; 364 aa; 14.6 kDa) is related to the Rep (ORF AC1) of *Begomovirus*. Pairwise identities showed that the highest OEGV-PT Rep identities (>50%) were found with those of the genera *Begomovirus*, *Curtovirus*, *Turncurtovirus*, *Topocuvirus*, which cluster together with OEGV-PT (I), as revealed by the phylogenetic analysis of the Rep of geminiviruses (Figure 2B). Four other clusters are composed by genera that present lower Rep identities to OEGV-PT; one cluster contains *Capulavirus* (II); other *Mastrevirus* (III); other *Grablovirus* (IV) and the other *Becurtovirus* and *Citlodavirus* and the unassigned MMDAV0 (V).

The translation product of TrAP (AC2) (459 nt; 152 aa; 17.7 kDa) is related to the TrAP of the *Begomovirus*. Pairwise identities showed that the highest OEGV-PT TrAP identities (>35%) were found with the NW *Begomovirus*, followed by OW *Begomovirus* and *Maldovirus* (>29%). The lowest OEGV-PT TrAP identities (<22%) were found with the *Becurtovirus*, *Grablovirus* and *Mastrevirus*. The translation product of replication enhancer protein (REn), ORF AC3, (441 nt; 146 aa; 18.3 kDa) is also related to the REn (AC3) of the *Begomovirus*. Pairwise identities showed that the highest OEGV-PT REn identities (>30%) were found with the genera *Begomovirus*, *Turncurtovirus* and *Topocuvirus*. The lowest OEGV-PT Ren identities (<22%) were found with the genera *Grablovirus* and *Capulavirus*.

Blastp analysis showed that the translation product of ORF AV1 (768 nt; 255 aa; 29.9 kDa) is most similar to the coat protein (CP) of the *Mastrevirus* TYDV-A (31% identity; query coverage, 92%; E value, 4 × 10^−14^). Pairwise identities showed that the highest OEGV-PT CP identities were found with the genera *Mastrevirus*, *Curtovirus*, *Turncurtovirus* and *Becurtovirus*, which cluster together with OEGV-PT (cluster II), as revealed by the phylogenetic analysis of the CP of geminiviruses (Figure 2C). Two other clusters are composed by genera that present lower CP identities to OEGV-PT; one cluster contains *Begomovirus*, *Capulavirus* and *Citlodavirus* and the unassigned MMDaV0 (cluster I) and the other cluster contains *Grablovirus* and *Maldovirus* (cluster III).

Multiple alignment of the OEGV-PT Rep protein sequences showed that OEGV-PT has sequence motifs typical of geminivirus (Appendix A). In addition, pairwise analysis of OEGV-PT Rep showed higher similarities to NW *Begomovirus* and SLCV clade (Appendix A). Interestingly, the alignment of Rep aa sequences showed, at the motif III, some aa homologies with the unique signatures of SLCV clade (Appendix A).

The OEGV-PT CP aa sequence displays low conserved amino acids between the geminivirus compared to Rep, however, it was interesting to verify several conserved amino acids regions (CR). Motif PWRsMaGT, conserved in the NW *Begomovirus* [44] is not present in the OEGV-PT. In addition, we identified several conserved regions (CR) within the geminivirus CP (CR I to CR VI) (Figure 3). These regions show many similarities with *Mastrevirus* corroborating our phylogenetic analysis and pairwise comparisons. Among these, CR I is conserved between OEGV-PT and the *Mastrevirus* (MSV) among other genera; CR III or putative motif candidates R (Rxx) presents the amino acid R, at position 132, which is conserved among geminivirus and the sequence RHT is conserved between OEGV-PT, *Mastrevirus* and *Maldovirus*. OEGV-PT also seems to be more similar to *Mastrevirus* and other genera in the CR V region, where it is clearly distinct from *Begomovirus*, that do not present this CP region. In addition, the CR VI region seems to be conserved among most geminiviruses used in this work, which may also indicate a putative motif candidates (ALY).

The BV1/MP of OEGV-PT DNA-B, showed the absence of the putative tyrosine phosphorylation site [RK]-x(2,3)-[DE]-x(2,3)-Y, a site typically present in the NW *Begomovirus* and absent in OW *Begomovirus*. However, the comparison of the putative tyrosine phosphorylation site in the NW *Begomovirus* MP with the homologous region in OW *Begomovirus* MP and OEGV-PT MP showed that only one aa substitution (corresponding to a single nt substitution) would be needed to change the functionality of this site to the tyrosine phosphorylation site found in the NW *Begomovirus* MP (Figure 4).

### 3.5. Recombination Analysis

The RDP analysis identified two recombination events for OEGV-PT (Figure 5), both well supported by at least five methods in the RDP4 package. One event (event 1) showed a recombination with the beginning breakpoint at nucleotide 141 (IR) and ending breakpoint at nucleotide 1506 (Rep and TrAP), involving a segment of 1365 nt long, encompassing the complete putative CP last, the complete C3, the last 433 nt of the TrAP and the last 63 nt of the putative OEGV Rep. This segment was putatively derived from an unknown minor parent and from the NW *Begomovirus* BGYMV (major parent). A second event (event 2) showed a recombination with the beginning breakpoint at nucleotide 2258 (Rep) and ending breakpoint at nucleotide 2720 (IR), involving a segment of 462 nt long, encompassing the first 259 nt of the putative OEGV Rep. This segment was putatively derived from ICMV (minor parent) and ACMV (major parent), both OW *Begomovirus*. No clear evidence for recombination was found in DNA-B.

### 3.6. Specificity-Determining Positions (SDPs) in CP Amino Acid Sequences

The automatic subgrouping allowed us to divide geminiviruses into eight groups that share the same type of insect vector (Table 2). SDP analysis did not cluster OEGV-PT together with any of the viral groups with known vectors.

## 4. Discussion

The advent of HTS has greatly increased the rate of plant virus discovery, from ~900 species in 2005 to 9110 species in 2021. However not all newly discovered plant viruses induce disease symptoms and their biological relevance presents a major challenge to plant pathologists and policy makers. Whilst studies on plant virus identification by HTS are dominated by herbaceous crops [45,46], those analyzing woody crops are less prevalent, probably due to the difficulty of obtaining high quality RNA. However, grapevines and citrus trees are an exception, most likely due to their economic importance [47,48,49].

In this work we performed HTS in olive and we identified OEGV in Portugal (OEGV-PT). The initial sequence results enabled us to sequence the full genome of the bipartite OEGV-PT. The OEGV-PT genome is identical, 100% in DNA-A and 99% in DNA-B, to the recently described OEGV [8], but further genomic and phylogenetic analyses performed in this work show that OEGV presents genomic characteristics from both OW and NW *Begomovirus*, different to presented previously [8]. These results were possible through a higher representativeness of geminivirus isolates in phylogenetic analysis, especially with the inclusion of NW *Begomovirus* from SLCV clade. Members of the NW *Begomovirus* SLCV lineage display features that distinguishes them from other NW begomoviruses [50], and in this study it was possible to identify several aa homologies in motif III in the N-terminal of OEGV-PT Rep which presents unique signatures of SLCV clade, and which may have contributed to the increased similarities with this lineage.

The length of both OEGV-PT DNAs is typical of the OW *Begomovirus* [51] but OEGV-PT genome organization is similar to NW *Begomovirus*, but no AC4/C4, typical of both NW and OW *Begomovirus*, was found. C4 is required for monopartite *Begomovirus* infection and for a few bipartite begomoviruses [52,53]. C4/AC4 is related to the induction of characteristic viral symptoms [54,55] and has been suggested to interact with CP and/or MP to transport the replicated genome from the nucleus to the cytoplasm and from cell-to-cell [56]. No begomoviruses lacking C4/AC4 have been described until now, however, other genera within the *Geminiviridae*: *Becurtovirus*, *Capulavirus*, *Grablovirus*, *Eragrovirus*, *Mastrevirus* and other species also lack C4, showing that in these viruses C4 is not essential for viral infection and function may be compensated by other viral proteins.

DNA-B components are only present in members of genus *Begomovirus* (mostly from NW and less from OW). OEGV-PT DNA-B comprises two ORFs: BC1/unknown, was found in the complementary sense and BV1/MP in the virion sense, in contrast to all other known bipartite begomoviruses, where MP is in complementary sense. In OEGV-PT MP, a single nt substitution would be needed to change this site into a functional tyrosine phosphorylation site, placing OEGV-PT DNA-B closer to NW *Begomovirus* and suggesting that OEGV-PT MP is under purifying selection. Tyrosine phosphorylation has shown to be involved in plant virus localization and cell to cell movement [57] and, in OW *Begomovirus*, the lack of the tyrosine phosphorylation function, may be compensated by AV2 [58,59]. It is, however, interesting to observe that OEGV-PT lacks AV2, and viruses that lack AV2, such as NW *Begomovirus*, must rely on DNA-B proteins for infectivity [20]. Our phylogenetic analyses showed important differences in OEGV-PT movement proteins in relation to all other geminiviruses. OEGV-PT proteins (including BC1) must have acquired novel roles during evolution, that allow host infection even in the absence of the AV2, AC4 and NSP. Further experiments are required to uncover the current functions of OEGV-PT proteins and understand the mechanisms of infection and movement within the host.

An additional clear distinct feature of OEGV-PT DNA-A and DNA-B is the length of its common region (CR), 403 nt, in opposition to the 348 nt recently referred by Chiumenti [8] for OEGV, which is, to our knowledge, the longest CR observed between DNA-A and its cognate DNA-B. In addition, it was interesting to verify that the OEGV-PT DNA-A and DNA-B CR contains the TATA box and four replication-associated iterons with a unique arrangement compared to the other known geminiviruses, and these sites are a major determinant of virus-specific replication [60,61], preventing Rep from replicating noncognate DNA-A and DNA-B components showing a very strong specificity between DNA-A and DNA-B and that reassortment in OEGV-PT is very unlikely to occur.

Our analysis of the OEGV-PT proteins showed that whilst other virus proteins were most closely related to the *Begomovirus*, OEGV-PT CP was more closely related to *Mastrevirus* and *Curtovirus* (Figure 3C). The low sequence identity of the OEGV-PT CP with other known genera, suggest that this protein presents functions that may be a mixture from different geminiviruses. For example, it is interesting to note that in bipartite *Begomovirus*, nucleocytoplasmic transport is provided by NSP and in monopartite viruses this function is fulfilled by the CP [62,63]; this may be the case in OEGV-PT, as it lacks a typical NSP. In addition to that and despite the high diversity between the OEGV-PT CP and the CP from other members of the family *Geminiviridae*, it is possible to suggest, based on the amino acid CP alignment, two new putative motif candidates (CR III or motif R and CR VI or motif ALY) among the geminiviruses, although functional studies with OEGV must be carried out to confirm the existence of new motifs in the family *Geminiviridae* or among some genera.

Among several roles of the viral CP, such as viral movement and infectivity [64,65,66,67], the CP is known to be involved in vector transmission and specificity [68,69,70]. Interestingly, *Mastrevirus* and *Curtovirus* are transmitted by leafhoppers, in opposition to the whitefly vectors of begomoviruses. It has been shown that CP of geminiviruses that share an insect vector cluster together in phylogenetic analyses. Here, we showed that OEGV-PT CP clustered together with *Mastrevirus*, *Turncurtovirus*, *Curtovirus* and *Becurtovirus*, which are transmitted by leafhoppers, suggesting that OEGV-PT CP similarities may reflect shared vectors. However, further clustering analysis suggested that the OEGV-PT vector, if any, may be different from those described for known geminiviruses. Further experiments are needed to gain information on the OEGV-PT vector. The low similarity of the OEGV-PT CP with begomoviruses CPs, may be the result of an adaptation of an ancient *Begomovirus* which lost the capacity to be transmitted by the whitefly *Bemisia tabaci*, an insect that is not frequently found in olive, and adapted for a more successful dissemination in this new host.

There are an increasing number of studies that show that bipartite begomoviruses, typical of NW are found in the OW and monopartite begomoviruses, typical of OW are being found in the NW [21,71,72]. Our data support the conclusion that the evolution of bipartite begomoviruses occurred following the appearance of monopartite species which acquired a DNA-B component, probably derived from the original DNA-A. Although the OEGV-PT CP is more similar to those of *Mastrevirus* and *Curtovirus*, our search of possible recombination events in OEGV-PT DNA-A did not show any recombination with these genera, instead, two recombination events for OEGV-PT, both involving begomoviruses were detected, suggesting that OEGV-PT evolved from begomoviruses. Due to the fact that no other geminiviruses have been found in olive, we suggest that OEGV-PT evolution occurred in a different host and may have started with an ancient OW monopartite *Begomovirus* that lost V2 and C4, gaining functions on cell to cell movement by acquiring a DNA-B component. The invasion of new host cells may have facilitated its acquisition by a new polyphagous vector that mediated its introduction into olive. Intensive surveys that include sampling of wild relatives and weed species near the infected orchards would help to gain insight into the origins of OEGV-PT.

Our characterization of OEGV-PT, highlighting for the first time the similarities and differences to both OW and NW begomoviruses as well as *Mastreviruses* and *Curtoviruses*, clearly shows that this virus represents a new evolutionary direction in the *Geminiviridae*. HTS allows the identification of asymptomatic novel geminiviruses, as is the case of OEGV-PT, and it is possible that evolution is favoring less pathogenic variants. Alternatively, such viruses may have alternative hosts that function as reservoirs, from which they are transmitted to new hosts, where they cause emergent and sometimes devastating diseases.

As more geminiviruses are discovered, there is a clear need to better understand their evolutionary traits and host impact. Whilst OEGV-PT is currently asymptomatic in olive, studies concerning its cellular location, symptoms, transmission, are essential to elucidate host–virus interactions and prevent it becoming an emergent disease in an alternative economically important host.

## Figures and Tables

**Figure 1 viruses-13-02035-f001:**
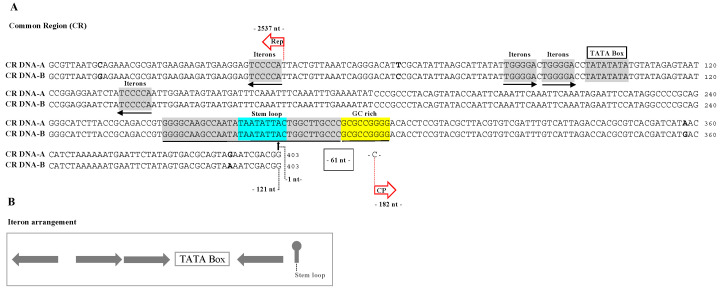
(**A**) Common region of OEGV-PT DNA-A and DNA-B alignment (5′–3′). Iterons and complementary iterons sequences are shown in gray, black arrows indicate their relative orientation; TATA box is shown in gray; the nonanucleotide sequence is shown in blue together with the sequences with the potential to form a stem-loop structure (in gray) to the right and to the left. The differences in the nucleotide sequences between both OEGV-PT DNAs (A and B) are shown in bold. The ↓ (black arrow) indicates position 1 in the viral genome corresponding to the predicted replication origin of the viral DNA. GC rich region in shown in yellow. (**B**) Iterons arrangement in OEGV-PT DNA-A and DNA-B alignment (5′–3′) positioned in relation to the TATA box and stem loop. Gray arrows indicate two iterons (virion-sense) and two complementary iterons (complement-sense).

**Figure 2 viruses-13-02035-f002:**
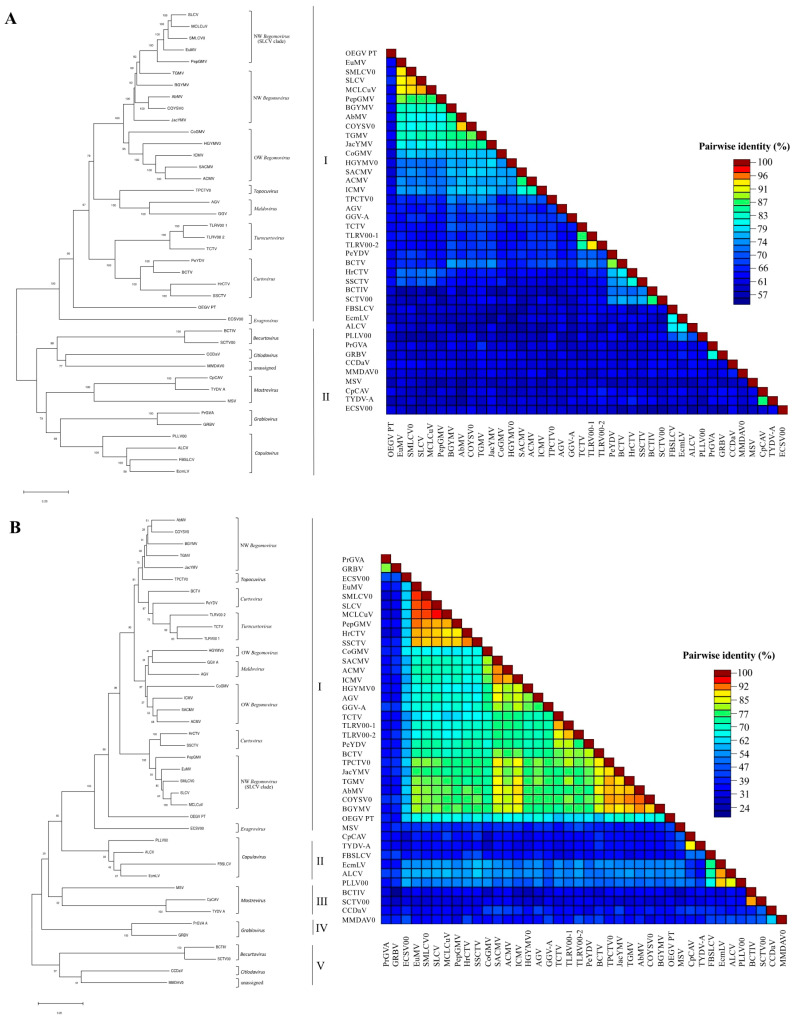
Phylogenetic trees constructed by the neighbor-joining (NJ) analysis from the alignment of the (**A**) OEGV PT full DNA-A (nt), (**B**) AC1/Rep (aa) and (**C**) AV1/CP (aa) from this study plus 39 sequences, with most representative geminiviruses including some isolates that had not yet been classified into genera, retrieved from the GenBank database. Virus genera, species and abbreviations, and GenBank accession numbers used in the NJ analysis are NW *Begomovirus* (SLCV clade); EuMV, SMLCV0, SLCV, PepGMV, MCLCuV, NW *Begomovirus*; AbMV, COYSV0, TGMV, BGYMV, JacYMV, OW Begomovirus; CoGMV, HGYMV0, SACMV, ACMV, ICMV, *Mastrevirus*; MSV, CpCAV, TYDV-A, *Topocuvirus*; TPCTV0, *Turncurtovirus*; TCTV, TLRV00-1, TLRV00-2, *Curtovirus*; PeYDV, BCTV, HrCTV, SSCTV, *Eragrovirus*; ECSV00, *Capulavirus*; FBSLCV, EcmLV, ALCV, PLLV00, *Becurtovirus*; BCTIV, SCTV00, *Grablovirus*; PrGV-A, GRBV, *Citlodavirus*; CCDaV, *Maldovirus*; AGV, GGV-A, and unassigned MMDAV0. Multiple sequence alignments were generated using MEGA 7 and the neighbor joining (BioNJ algorithms), based on calculations from pairwise nucleotide (nt) and amino acid (aa) sequences distances for full genome (nt) or proteins (aa) analysis. Numbers above the lines indicate bootstrap scores out of 1000 replicates. Roman numbers indicate the groups clustered in each phylogenetic tree.

**Figure 3 viruses-13-02035-f003:**
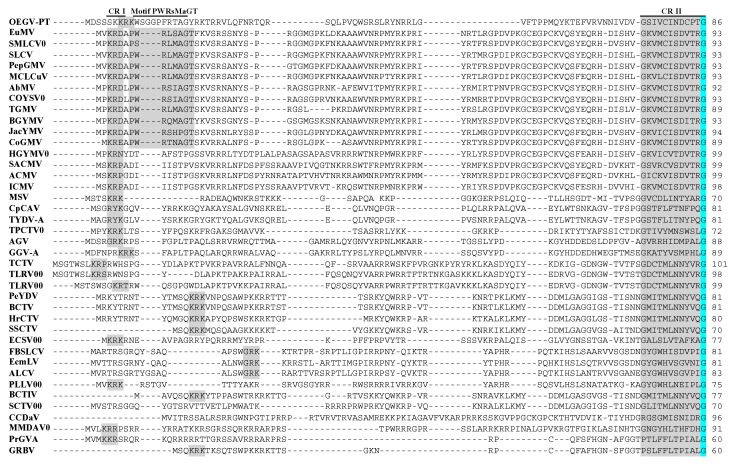
Multiple alignment of OEGV-PT CP protein sequences with most representative geminiviruses including some isolates that had not yet been classified into genera, retrieved from the GenBank database. Gray boxes are highlighting several conserved regions (CR) or motifs, and blue indicate 100% homology between the geminivirus. Conserved regions (CR) or motifs; CR I, motif PWRsMaGT (conserved on NW Begomovirus), CR II, CR III or putative motif candidates R, CR IV, CR V (amino acids enriched region in OEGV-PT) and CR VI or putative motif candidates ALY. Virus species and genera are NW *Begomovirus* (SLCV clade); EuMV, SMLCV0, SLCV, PepGMV, MCLCuV, NW *Begomovirus*; AbMV, COYSV0, TGMV, BGYMV, JacYMV, OW *Begomovirus*; CoGMV, HGYMV0, SACMV, ACMV, ICMV, Mastrevirus; MSV, CpCAV, TYDV-A, *Topocuvirus*; TPCTV0, *Turncurtovirus*; TCTV, TLRV00-1, TLRV00-2, *Curtovirus*; PeYDV, BCTV, HrCTV, SSCTV, *Eragrovirus*; ECSV00, *Capulavirus*; FBSLCV, EcmLV, ALCV, PLLV00, *Becurtovirus*; BCTIV, SCTV00, *Grablovirus*; PrGV-A, GRBV, *Citlodavirus*; CCDaV, *Maldovirus*; AGV, GGV-A, and unassigned genera; MMDaV.

**Figure 4 viruses-13-02035-f004:**
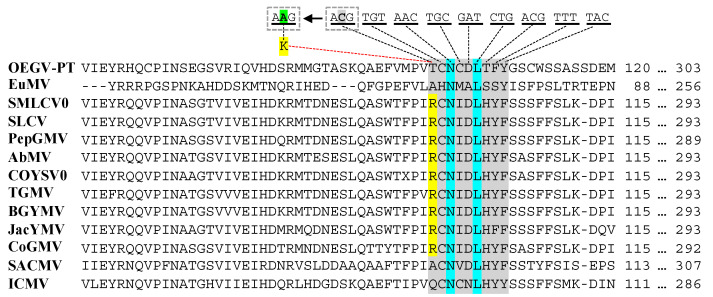
Alignment of OEGV-PT DNA-B MP aa sequences with begomoviruses showing (in gray) the putative tyrosine phosphorylation site [RK]-x(2,3)-[DE]-x(2,3)-Y in the NW *Begomovirus* and the homologous OW and OEGV-PT region. On top, codons of OEGV-PT sequence are shown. In green is shown how a single nt substitution (C by A) in the first codon of the site would be enough to change the functionality of tyrosine phosphorylation (T by K) in OEGV-PT. Blue shaded aa represent conserved aa among begomoviruses.

**Figure 5 viruses-13-02035-f005:**
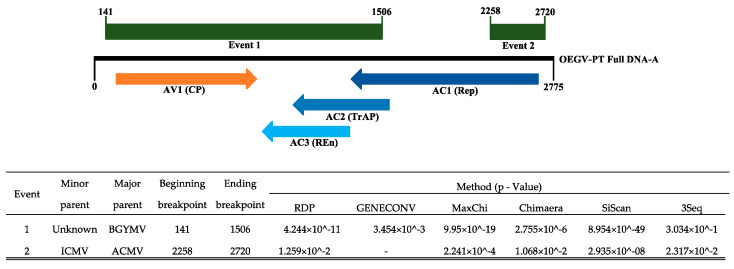
Schematic representation of the recombination events detected by RDP4.101 in OEGV-PT genome and corresponding p-values. A simplified linearized map was used to represent OEGV-PT full DNA-A. The floating boxes above the genome show the position of the two recombination events in OEGV-PT DNA-A. The floating arrows below the genome correspond to the position of each of the putative OEGV genes.

**Table 1 viruses-13-02035-t001:** Summary statistics of (1) raw sequence read pairs, (2) trimming, (3, 4) host mapping and (5, 6) contig assembly.

		(1)	(2)	(3)	(4)	(5)	(6)
Sites	Sample ID	Raw	Trimmed	Nonhost	Host	>200 bp	>2500 bp
		Reads	Reads	Reads	reAds (%)	Contigs	Contigs
Site A	GM1	1,439,571	1,354,917	718,728	47. 0	3.433	1
	CM2	1,829,542	1,734,121	943,211	45. 6	2.803	1
	PM3	882,313	833,859	479,347	42. 5	743	1
	AM4	1,218,856	1,125,962	621,743	44. 8	11.968	5
Site B	GF13	1,234,906	1,166,501	632,432	45. 8	4.572	10
	CF14	1,567,086	1,480,478	786,043	46. 9	5.106	8
	PF15	1,267,441	1,197,144	670,079	44. 0	4.759	1
	AF16	1,612,115	1,527,259	771,328	49. 5	14.093	39
Site C	GL9	1,744,076	1,644,553	897,574	45. 4	8.116	23
	CL10	1,022,113	953,274	533,379	44. 0	529	3
	PL11	1,897,805	1,804,294	883,768	51. 0	21.911	67
	AL12	1,461,607	1,376,263	733,076	46. 7	3.404	0
Site D	GS5	825,863	750,849	43,448	42. 1	3.426	2
	CS6	2,073,509	1,974,815	1,074,547	45. 6	7.887	5
	PS7	1,796,210	1,699,435	914,666	46. 2	2.667	1
	AS8	1,311,804	1,249,506	675,575	45. 9	1.748	1

**Table 2 viruses-13-02035-t002:** Specificity-determining positions (SDPs) alignment for the CP amino acid sequences of the geminivirirus. CP multiple sequence alignment by clustalX code. The sequences were automatically divided into eight groups by SCI-PHY and SDPs calculated by the SPEER server. The columns indicate the groups, genera, virus species (acronym), known insect vectors, the aa predicted to determine differences in virus vector specificity and respective positions in the CP multiple alignments. The gray boxes indicate the homologies in the specific aa.

Groups	Genera	Species	Vectors	Amino Acids and Positions								
				102	120	137	144	244	247	253	256	268
1	*Citlodavirus*	CCDaV	unknown	M	I	N	N	F	H	V	T	S
	unassigned	MMDAV0		H	I	N	H	Y	H	L	V	S
2	*Maldovirus*	AGV		H	I	Q	V	Q	N	W	L	V
		GGV-A		Y	I	N	I	G	D	Y	I	I
3	*Grablovirus*	PrGV-A		F	L	V	H	K	D	R	V	A
		GRBV	*S. festinus*	F	L	V	H	K	D	R	V	A
4	*Eragrovirus*	ECSV00	leafhopper	L	T	T	V	W	S	I	I	R
	*Mastrevirus*	MSV		L	T	S	G	W	N	V	I	G
		CpCAV		L	T	M	F	W	N	V	I	A
		TYDV-A		L	T	M	F	W	N	V	I	G
5	*Capulavirus*	EcmLV	*A. tirucallis*	H	I	A	Y	F	S	Y	T	Q
		ALCV	*A. craccivora*	H	I	A	W	F	S	Y	T	Q
		FBSLCV	unknown	H	I	Q	W	F	S	Y	T	Q
		PLLV00		H	V	G	V	F	S	Y	T	Q
6	*Turncurtovirus*	TCTV	*C. haematoceps*	M	T	F	M	W	D	Y	L	D
		TLRV00-1		M	T	F	M	W	D	Y	L	D
		TLRV00-2		M	T	F	M	W	D	Y	L	D
7	*Curtovirus*	PeYDV	*C. tenellus*	M	T	F	M	W	D	Y	V	D
		BCTV		M	T	F	M	W	D	Y	V	D
		HrCTV	unknown	M	T	F	M	W	D	Y	V	D
		SSCTV		M	T	F	M	W	D	Y	V	D
	*Becurtovirus*	SCTV00		M	T	F	M	W	D	Y	V	D
		BCTIV	*C. haematoceps*	M	T	F	M	W	D	Y	V	D
8	*Begomovirus*	EuMV	*B. tabaci*	C	F	I	N	Y	H	Y	H	T
		SMLCV0		C	F	I	N	Y	H	Y	H	T
		SLCV		C	F	I	N	Y	H	Y	H	T
		PepGMV		C	F	I	N	Y	H	Y	H	T
		MCLCuV		C	F	I	N	Y	H	Y	H	T
		AbMV		C	F	I	N	Y	H	Y	H	T
		COYSV0		C	F	I	N	Y	H	Y	H	T
		TGMV		C	F	I	N	Y	H	Y	H	T
		BGYMV		C	F	I	N	Y	H	Y	H	T
		JacYMV		C	F	I	N	Y	H	Y	H	T
		CoGMV		C	F	I	N	Y	H	Y	H	T
		HGYMV0		C	F	I	N	Y	H	Y	H	T
		SACMV		C	F	I	N	Y	H	Y	H	T
		ACMV		V	F	I	N	Y	H	Y	H	T
		ICMV		C	F	I	N	Y	Q	Y	H	T
-	unassigned	OEGV-PT	unknown	C	V	A	Q	F	D	V	M	F
-	*Topocuvirus*	TPCTV0		Y	E	A	Y	Y	D	I	I	D

## Data Availability

The datasets presented in this study can be found in online repositories. The names of the repositories and accession numbers of both DNA-A and DNA-B can be found below: https://www.ncbi.nlm.nih.gov/genbank/, accessed on 2 September 2021, MZ355666, MZ355667, respectively.

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
