# Peer review of "A Bipartite Geminivirus with a Highly Divergent Genomic Organization Identified in Olive Trees May Represent a Novel Evolutionary Direction in the Family Geminiviridae"

_viruses, 2021, doi:10.3390/v13102035_

Round 1

Reviewer 1 Report

I thank the authors for putting together this manuscript. It is an interesting paper describing a putative new isolate of Olea Europa geminivirus (OEGV) obtained in olive in Portugal. The manuscript is well written and presented but I have some questions and concerns about the novelty of some of the results described in this manuscript. Therefore, my recommendation is for authors to address the queries below and resubmit the manuscript to the journal.

My major concern is that some of the results presented in this manuscript, namely the ones in sections 3.2; 3.3; 3.4 regarding genome characterization are the same described by Chiumenti et al. 2021. I will highlight these below in my comments and questions in each section of the manuscript.

Introduction:

Introduction is good but needs some rephrasing. my comments below:

line 51: I would recommend rephrasing it because Geminiviridae is one of the largest family of plant virus alongside the Potiviridae.

line 53: the word “small” can be removed.

lines 62-64: please rephrase it. and what do authors mean by “most ecologically successful group”?

lines 71-79: this paragraph on the genome organization of begomovirus between old world and new world is confusing. Please rephrase it. Also, line 78 says “OW Begomovirus are monopartite”; does this contradicts line 22 in the abstract?

Materials and Methods:

this section is well described. I just have a couple of comments:

line 142: please include the storage conditions of the cortical scrapings

line 151: please define EACC

Results:

As I said above, results are overall well presented and described. However, in my view a lot of the results have already been described before by Chiumenti et al. 2021.

line 241: “a number of PFAM7 conserved domains”; this sentence is vague and needs rephrasing

lines 245-247: please include the fragment size of DNA-A and DNA-B of OEGV to get a better understanding if the contigs 2727bp and 1286bp are close or not to full genome sequences. Also, could authors discuss the 100% identity with OEGV? Is this expected from different virus isolates?

line 251: what do you mean by “divergent contigs”?

line 256: “2775bp and 2763 bp”: these genome sizes are exactly the same size as OEGV. is this expected?

line 263: please include what robust test would you use to validate the specificity of OEGV-PT to Galega vulgar.

line 264: this section 3.3 is very similar to the results presented by Chiumenti et al. 2021. there is no novelty in this description. I would suggest to edit this section completely. figure 1 is also very similar to Chiumenti et al. 2021.

Line 305: beginning of this section (first two paragraphs) same as reported by Chiumenti et al. 2021.

Line 322-324: could this SNP be the result of sequencing error? did authors checked this with other methods, e.g. sanger sequencing?

Line 358: Figure 3 – please include the different genera on the trees to make it easier to understand.

line 436: section on “Prediction of insect vector through CP analysis”. This is very interesting but also very speculative without any biological data to confirm the results. As an example, Ateka et al. 2017 reported that a DAG motif associated with aphid-transmitted viruses was found on Cassava brown streak virus (CBSV) and that it suggested that aphids could be a vector of CBSV. However, till today no aphids were reported to transmit CBSV and whiteflies remain the main vector.

I would recommend authors to remove this section or to add more data to make this section less speculative.

Line 558: supplementary material does not include the original OEGV for comparison! Could you please explain why? I would recommend adding OEGV to the analysis.

Author Response

Comments and Suggestions for Authors

Reviewer #1

Comments and Suggestions for Authors

I thank the authors for putting together this manuscript. It is an interesting paper describing a putative new isolate of Olea Europa geminivirus (OEGV) obtained in olive in Portugal. The manuscript is well written and presented but I have some questions and concerns about the novelty of some of the results described in this manuscript. Therefore, my recommendation is for authors to address the queries below and resubmit the manuscript to the journal.

Comment: My major concern is that some of the results presented in this manuscript, namely the ones in sections 3.2; 3.3; 3.4 regarding genome characterization are the same described by Chiumenti et al. 2021. I will highlight these below in my comments and questions in each section of the manuscript.

Reply: Our initial idea of keeping some of the information that was described by Chiumenti et al. 2021 was to make our paper more easy to read, gathering all the information, however, we understand the concern of reviewer #1 and we have made all the changes requested by reviewer #1.

Introduction:

Introduction is good but needs some rephrasing. my comments below:

Comment: line 51: I would recommend rephrasing it because Geminiviridae is one of the largest family of plant virus alongside the Potiviridae.

Reply: We agree with the reviewer’s comments and we rephrased the information. We included the information: “and belongs to the family Geminiviridae that is the largest family of plant viruses alongside the family Potyviridae. Members from the family Geminiviridae are present in tropical and subtropical regions and constitute over 520 distinct species.”

Comment: line 53: the word “small” can be removed.

Reply: The word was removed as suggested.

Comment: lines 62-64: please rephrase it. and what do authors mean by “most ecologically successful group”?

Reply: We agree with the reviewer that the statement “most ecologically successful group” is very vague, for this reason we believe now the sentence is simpler: “Genera Begomovirus and Mastrevirus correspond to 94% of the Geminiviridae virus species, although Begomovirus is by far the genus with the highest number of recognized species, over 445 [16]. This remarkable success, in the higher number of species,...”

Comment: lines 71-79: this paragraph on the genome organization of begomovirus between old world and new world is confusing. Please rephrase it.

Reply: We changed the sentence and believe it is now more clear “The DNA-A of bipartite NW begomoviruses has five open reading frames (ORFs), one in the virion-sense encoding the coat protein (AV1 or CP) and four in the complementary sense encoding the replication initiation protein (AC1 or Rep), the transcriptional activator (AC2 or TrAP), the replication enhancer (AC3 or REn) and the AC4 protein (or C4) [23,24]. The DNA-B of bipartite NW begomoviruses comprises two ORFs; one in the virion-sense encoding the nuclear shuttle protein (BV1 or NSP); and one in the complementary sense encoding the movement protein (BC1 or MP) [19,25].”

Also, line 78 says “OW Begomovirus are monopartite”; does this contradicts line 22 in the abstract?

Reply: As suggested by the reviewer we clarified the sentence in the introduction: “In contrast with the NW begomoviruses, the majority of OW begomoviruses are monopartite with their DNA comprising 6 ORFs, presenting an additional ORF V2 or AV2, also referred as pre-coat [26], and that is not present in NW begomoviruses.”

And In the abstract: “The bipartite genome (DNA-A and DNA-B) of the OEGV-PT is similar to Old World begomoviruses inlength, but it lacks a pre-coat protein (AV2), which is a typical feature of New World begomoviruses (NW).”

Materials and Methods:

this section is well described. I just have a couple of comments:

Comment: line 142: please include the storage conditions of the cortical scrapings

Reply: The information was included as suggested by the reviewer: “Cortical scrapings of the 3-year stems of each sampled tree were ground in liquid nitrogen, mixed in equal amounts, combined per cultivar at each site (4 cultivars x 4 sites) and stored at −80 °C.”

Comment: line 151: please define EACC

Reply: The sentence was improved as suggested by the reviewer: “The controls included an uninfected tobacco leaf spiked with External RNA Control Consortium (ERCC) positive control artificial RNA.”

Results:

As I said above, results are overall well presented and described. However, in my view a lot of the results have already been described before by Chiumenti et al. 2021.

Comment: line 241: “a number of PFAM7 conserved domains”; this sentence is vague and needs rephrasing

Reply: When we did the BlastP search against the Genbank virus genomes with the 2727bp and 1286bp length contigs, the sequences from the paper Chiumenti et al. 2021 were not yet available in the database, but when our manuscript was submitted to Viruses the sequences of Chiumenti et al. 2021 were already available, therefore, we changed this sentence.

The new sentence is: “A BlastP search against the Genbank virus genomes with the 2727bp and 1286bp length contigs, showed 100% identity in DNA-A (accession number MW316657; 2775bp in length) and 99.96% identity in DNA-B (MW316658; 2763bp in length), which both correspond to a geminivirus found in olive in Italy and named Olea europaea Geminivirus (OEGV) [8].”

Comment: lines 245-247: please include the fragment size of DNA-A and DNA-B of OEGV to get a better understanding if the contigs 2727bp and 1286bp are close or not to full genome sequences. Also, could authors discuss the 100% identity with OEGV? Is this expected from different virus isolates?

Reply: The information were included and, the paragraph was improved as suggested by the reviewer. Please see the previous comment.

Comment: line 251: what do you mean by “divergent contigs”?

Reply: We changed the word “divergent” to “different”.

Comment: line 256: “2775bp and 2763 bp”: these genome sizes are exactly the same size as OEGV. is this expected?

Reply:  OEGV-PT genome has a high similarity with the one described by Chiumenti et al., 2021, so it might be anticipated that they might also have the same lengths.

The new sentence: “The full nucleotide (nt) sequencing of both contigs (2727bp and 1286bp) revealed two circular DNAs; of 2775bp and 2763 bp respectively, presenting the same length of OEGV genomes described by Chiumenti et al. [8].”

Comment: line 263: please include what robust test would you use to validate the specificity of OEGV-PT to Galega vulgar.

Reply: It would be interesting in a future work to increase the number of olive trees of other cultivars (Cobrançosa, Picual and Arbequina), as well as the number of Portuguese regions (or countries) and the number of trees (replicates), to ensure that in fact this virus is not present in these cultivars. But it's interesting to say that we sampled the four cultivars in the four different sites, and in each site they have the same mode of management (for example, automatic pruning and harvesting), that could facilitate the transmission of this virus in the different cultivars (from tree to tree), however it was not the case.

The sentence was improved as suggested by the reviewer: “All trees at site A were negative. An intriguing result was that all positive trees belonged to the cultivar Galega vulgar, suggesting a certain specifity of OEGV-PT for this cultivar, however more robust sampling, in terms of number of trees, fields, or regions is needed to validate this observation.”

Comment: line 264: this section 3.3 is very similar to the results presented by Chiumenti et al. 2021. there is no novelty in this description. I would suggest to edit this section completely. figure 1 is also very similar to Chiumenti et al. 2021.

Reply: We agree with the reviewer that some information has already been showed by Chiumenti et al. 2021, as we already explained in a previous comment. We have made changes throughout this section as suggested. We only maintained the description of the common region which is very different from the analyses performed by Chiumenti.

The new sentence: “The nucleotide (nt) similarity, size and genome organization of OEGV-PT DNA-A is identical to OEGV isolate from Italy (Chiumenti et al. 2021). Interestingly, and also as verified by Chiumenti et al. (2021) no C4, typical of the begomoviruses, was found. The full nucleotide (nt) sequence of OEGV-PT DNA-A (2775bp) presents a length typical of OW begomoviruses and an organization similar to NW begomoviruses. The DNA-B of OEGV-PT presents the same length (2763 bp) and genome organization of OEGV isolate from Italy (Chiumenti et al. 2021) but differs in a single nucleotide within the BV1/MP.

In addition the figure 1 (now Figure S1) was included in the supplementary files.

Comment: Line 305: beginning of this section (first two paragraphs) same as reported by Chiumenti et al. 2021.

Reply: We changed both paragraphs as suggested by the reviewer. “OEGV-PT DNA-A pairwise and phylogenetic analysis of AC1 and AV1 predicted proteins confirms divergent results in terms of geminiviral origin as suggested by Chiumenti et al. [8] (Figure 3). We included the analysis of AC2 and AC3 proteins, both showing to be more related to Begomovirus, as verified for AC1 (Table S3). We also included, in this analysis, a new group of NW begomoviruses, belonging to the SLCV clade with which OEGV-PT DNA-A showed the highest similarity (62.1%).

Comment: Line 322-324: could this SNP be the result of sequencing error? did authors checked this with other methods, e.g. sanger sequencing?

Reply: Yes, we confirmed and it is not an amplification nor a sequencing error. We use sanger sequencing in both directions (5’ to 3’ and 3’ to 5’).

The new sentence: “OEGV-PT full DNA-B sequence is 99.96% similar to OEGV full DNA-B from Italy (Accession number MW316658), with a single difference at nt 981, C in the Italian and T in the PT isolate.”

Comment: Line 358: Figure 3 – please include the different genera on the trees to make it easier to understand.

Reply: The genera were included as suggested by the reviewer.

Comment: line 436: section on “Prediction of insect vector through CP analysis”. This is very interesting but also very speculative without any biological data to confirm the results. As an example, Ateka et al. 2017 reported that a DAG motif associated with aphid-transmitted viruses was found on Cassava brown streak virus (CBSV) and that it suggested that aphids could be a vector of CBSV. However, till today no aphids were reported to transmit CBSV and whiteflies remain the main vector.

I would recommend authors to remove this section or to add more data to make this section less speculative.

Reply: We found interesting to present this information because it is well known the role of the viral CP in vector transmission. OEGV-PT has a low similarity with begomovirus CP, a genus that is transmitted by the whitefly Bemisia tabaci, an insect that is not frequently found in olive, which may indicate another vector. Even those genera (Mastrevirus, Curtovirus, Turncurtovirus and Becurtovirus) that showed a slight higher similarity in the CP only showed values of 30%. In addition, we think that this analysis is important to support our indication of a possible evolution of OEGV from a Begomovirus that changed its CP or the capacity to be transmitted by the Begomivirus vector.

However, we understand the concerns of the reviewer and we made substantial changes in this section. Reviewer #2 also made some comments related to this section and we hope that we have addressed concerns of both reviewers.

We changed the subtitle to ‘Specifity-determining positions (SDPs) in CP amino acid sequences’ and made some changes in the text. The new sentence: “The automatic subgrouping allowed us to divide geminiviruses into 8 groups that share the same type of insect vector (Table 2). SDP analysis did not cluster OEGV-PT together with any of the viral groups with known vectors.”

Comment: Line 558: supplementary material does not include the original OEGV for comparison! Could you please explain why? I would recommend adding OEGV to the analysis.

Reply: We have now included the OEGV from Italy in the supplementary material.

Reviewer 2 Report

The manuscript "A bipartite geminivirus with a highly divergent genomic organization identified in olive trees may represent a new evolutionary direction in the family Geminiviridae" by Materatski et al presents a compelling case for a different evolutionary path for a bipartite begomovirus found in olive trees in Portugal. The Olea europeae Geminivirus presents a longer common region and an unusual arrangement of iterons. Furthermore, the authors suggest a new direction for geminivirus evolution providing evidence that OEGV-PT may have derived from an ancient begomovirus from the Old World that has lost some ORFs and gained movement functions acquiring a B component.

I will suggest to change the "new" in the title to "different" or "novel" because new may confuse that is a direction that has just happened, while here the authors refer to a hierarchy that has not been proposed before.

The introduction is thorough and well written and I enjoyed reading it. 

I have a curiosity question for the authors about the sampling. It has been shown (mostly in herbal plants) that geminiviruses tend to accumulate in the apex of the plant. Why did the authors avoid sampling apex areas of the plant and focused on cortical scrapings? Do the leaves closer to the meristem area are not useful to detect these viruses?

The authors looked at RNA sequences first and then traced them back to DNA and designed abutting primers to amplify the full genomic sequences of the detected begomovirus.

When the authors used the DNA to show the genomic features of the A and B components, they are not presenting the open reading frame that the virus detected in Italy presented. How is this sequence different than the Italian one? Were the authors unable to find the message from the putative ORF detected by the Italian group? Please address this situation.

It is extremely interesting that the authors found that the Galega vulgar cultivar showed positives while others did not. I agree with them that more research should be performed on this matter.

In lane 286 it may be better to write... are cognates components of a bipartite virus. Or even "genome", instead of DNA, which is very generic.

In this section, maybe an iteron analysis may have provided light in which type of Rep protein this virus is presenting, maybe it could be a good addition to the otherwise very stringent analysis of the genome. Like is the case of the PaLCuV, CLCuLV and ToLCV-IN, which all have a TGGGGA core iteron.

With the CP analysis and description of the "newly found" motif R, it seems like there are certainly similarities but the authors should express if these may have functions associated. Are there any suggestions of this in previous work? This finding is very interesting and it seems crucial to assign functional character to the motifs.

The recombination analysis looks also very interesting. However, there is no comments about the actual origin of the olive tree, which may support the idea of where the virus came from (as long as it is a primary host, of course).

Based on the CP analysis for transmission, would the authors suggest a possible insect to vector this virus? It should be an insect that feeds in olive trees. Another interesting route for research.

The sentence in the discussion about that not all the newly found plant viruses induce disease symptoms is very true, but furthermore, a well adapted virus shouldn't induce a lot of symptoms in its host, in a way that can survive there for long time and be transmitted, without killing the host. The problem in plant virology is when a virus finds a "new host", and the host has never been exposed to this virus, then is when epidemics happen. Here it seems the authors found a well adapted virus to olive trees. However, still is to see if this virus is transmitted to other plants and how.

The authors also comment on their finding that the OEGV-PT has characteristics of the OW and the NW begomoviruses, however, it has a higher phylogenetic relation to the SQLV clade, would the authors please elaborate on this situation? Why do they think this is? How this clade is different than the other NW viruses and what is know about its phylogenetic origin?

Very interesting manuscript.

Author Response

Comments and Suggestions for Authors

Reviewer 2

Comments and Suggestions for Authors

The manuscript "A bipartite geminivirus with a highly divergent genomic organization identified in olive trees may represent a new evolutionary direction in the family Geminiviridae" by Materatski et al presents a compelling case for a different evolutionary path for a bipartite begomovirus found in olive trees in Portugal. The Olea europeae Geminivirus presents a longer common region and an unusual arrangement of iterons. Furthermore, the authors suggest a new direction for geminivirus evolution providing evidence that OEGV-PT may have derived from an ancient begomovirus from the Old World that has lost some ORFs and gained movement functions acquiring a B component.

Comment: I will suggest to change the "new" in the title to "different" or "novel" because new may confuse that is a direction that has just happened, while here the authors refer to a hierarchy that has not been proposed before.

Reply: The word “new” in the title was changed by “novel” as suggested by the reviewer.

The new title: “A bipartite geminivirus with a highly divergent genomic organization identified in olive trees may represent a novel evolutionary direction in the family Geminiviridae”

The introduction is thorough and well written and I enjoyed reading it. 

Comment: I have a curiosity question for the authors about the sampling. It has been shown (mostly in herbal plants) that geminiviruses tend to accumulate in the apex of the plant. Why did the authors avoid sampling apex areas of the plant and focused on cortical scrapings? Do the leaves closer to the meristem area are not useful to detect these viruses?

Reply: Thanks for the question, which is very interesting. Olive has several components, especially on leaves, such as phenols and polyphenols that are inhibitors and compromise a good RNA extraction, this is actually the fact that NGS was only more recently applied to olive. The main reason for our choice to work with the company FERA science, was because they have experience in working with “more difficult plants” … We also decided to use cortical scrapings because most olive viruses (at the time all of RNA viruses) have a higher titre in this plant tissue. We have to confess that we weren't expecting to find DNA viruses, it was a nice surprise.

Comment: The authors looked at RNA sequences first and then traced them back to DNA and designed abutting primers to amplify the full genomic sequences of the detected begomovirus.

When the authors used the DNA to show the genomic features of the A and B components, they are not presenting the open reading frame that the virus detected in Italy presented. How is this sequence different than the Italian one? Were the authors unable to find the message from the putative ORF detected by the Italian group? Please address this situation.

Reply: We found the same ORFs that the Italian OEGV. We removed that information that was already described by Chiumenti et al. 2021 as suggested by reviewer #1 in section 3.3. The only differences, were in the analysis of the common region, that were not fully addressed by the Italian group. They considered the CR of 348 and we showed that the CR has in fact 403 because Chiumenti stopped in the first nt difference they found. We hope to have clarified this issue.

Comment: In lane 286 it may be better to write... are cognates components of a bipartite virus. Or even "genome", instead of DNA, which is very generic.

Reply: As suggested by the reviewer we changed the word “DNA” by "genome"

Comment: In this section, maybe an iteron analysis may have provided light in which type of Rep protein this virus is presenting, maybe it could be a good addition to the otherwise very stringent analysis of the genome. Like is the case of the PaLCuV, CLCuLV and ToLCV-IN, which all have a TGGGGA core iteron.

Reply: When we did the search for the iterons, the nucleotide sequences of iterons of each virus showed a high variability, among the genera of geminivirus.

In this sense, we had already checked (before the submission) the configuration of the CLCuLV and ToLCV iterons, and not the PaLCuV, that we only made now.

The reason why we did not include an alignment of the Iterons (or the Common regions) with viruses with the same iteron sequence, as the case of these 3 viruses cited by the reviewer, was because although they have the same TGGGGA core iteron, they have different number and different configurations compared to OEGV, which is the most interesting question in our point of view.

A comment about that, for ToLCV and PaLCuV we only found one iteron TGGGGA (position 2374 and 2628, respectively). The inverted iterons were not found, which left us wondering if the sequence TGGGGA is really the iterons of these viruses, as it would be unusual (only 1 iteron). In the case of CLCuLV the Iteron is TGGGGA, however with a different configuration compared to OEGV; two TGGGGA (position 2627 and 2636), TATA box (position 2648) and one inverted iteron (position 2657).

Comment: With the CP analysis and description of the "newly found" motif R, it seems like there are certainly similarities but the authors should express if these may have functions associated. Are there any suggestions of this in previous work? This finding is very interesting and it seems crucial to assign functional character to the motifs.

Reply: We agree with the reviewer that this information is not clear, the only place in the manuscript where we say these are possible motif candidates is in the discussion, because in fact, no functional studies (in our work or in others) have been performed in this matter. For this reason, we have made an effort to change other sections of the manuscript, where we now indicate to be “putative motif candidates”. In the results section we included the information “putative motif candidates” in the description of the motifs, and in the discussion section we also indicate that functional studies should be performed to confirm the presence of new motifs for this family or some genera. The new sentence: “In addition to that and despite the high diversity between the OEGV-PT CP and the CP from other members of the family Geminiviridae, it is possible to suggest, based on the amino acid CP alignment, two new putative motif candidates (CR III or motif R and CR VI or motif ALY) among the geminiviruses, although functional studies with OEGV must be carried out to confirm the existence of new motifs in the family Geminiviridae or among some genera.”

Comment: The recombination analysis looks also very interesting. However, there is no comments about the actual origin of the olive tree, which may support the idea of where the virus came from (as long as it is a primary host, of course).

Reply: Yes, it is very interesting. We did not include that information because we did not  describe the scenario of olive as a primary host, we suggest that OEGV may have been originated from a Begomovirus that acquired the capacity to be transmitted by another vector and then was transmitted to olive.

Comment: Based on the CP analysis for transmission, would the authors suggest a possible insect to vector this virus? It should be an insect that feeds in olive trees. Another interesting route for research.

Reply: Unfortunately, and due to the large variability found between amino acids among the geminivirus it would be too speculative to suggest possible vector insect. Although these analyses were important to support the idea that “the low similarity of the OEGV-PT CP with begomoviruses CPs, may be the result of an adaptation of an ancient Begomovirus which lost the capacity to be transmitted by the whitefly Bemisia tabaci, an insect that is not frequently found in olive, and adapted for a more successful dissemination in this new host.”

In addition, we modified some of this information in this section also suggested by reviewer#1. We hope to have fulfilled the requests of both reviewers.

The new sentence: “The automatic subgrouping allowed us to divide geminiviruses into 8 groups that share the same type of insect vector (Table 2). SDP analysis did not cluster OEGV-PT together with any of the viral groups with known vectors.”

Comment: The sentence in the discussion about that not all the newly found plant viruses induce disease symptoms is very true, but furthermore, a well adapted virus shouldn't induce a lot of symptoms in its host, in a way that can survive there for long time and be transmitted, without killing the host. The problem in plant virology is when a virus finds a "new host", and the host has never been exposed to this virus, then is when epidemics happen. Here it seems the authors found a well adapted virus to olive trees. However, still is to see if this virus is transmitted to other plants and how.

Reply: Yes, we totally agree. Despite only being detected now, it is very possible that this virus is not so new in olive and it is already well adapted. In fact, reports of NGS for viruses in olive just began in 2021.

Comment: The authors also comment on their finding that the OEGV-PT has characteristics of the OW and the NW begomoviruses, however, it has a higher phylogenetic relation to the SQLV clade, would the authors please elaborate on this situation? Why do they think this is? How this clade is different than the other NW viruses and what is know about its phylogenetic origin?

Reply: We made some changes in the results and discussion sections to elaborate this situation: Results; “In addition, pairwise analysis of OEGV-PT Rep showed higher similarities NW Begomovirus and SLCV clade (Table S3). Interestingly, the alignment of Rep aa sequences showed, at the motif III, some aa homologies with the unique signatures of SLCV clade (Figure S1).” Discussion; “These results were possible through a higher representativeness of geminivirus isolates in phylogenetic analysis, especially with the inclusion of NW Begomovirus from SLCV clade. Members of the NW Begomovirus SLCV lineage display features that distinguishes them from other NW begomoviruses (Torres-Herrera, et al., 2019), and in this study it was possible to identify several aa homologies in motif III in the N-terminal of OEGV-PT Rep which presents unique signatures of SLCV clade, and which may have contributed to the increased similarities with this lineage.”

Comment: Very interesting manuscript.

Reply: We appreciate the questions and suggestions, which we believe improved the quality of the manuscript.

Round 2

Reviewer 1 Report

Thank you Authors for addressing all reviewers comments. It is a very interesting manuscript that will catch a lot of attention.

I recommend this manuscript to be accepted for publication. Congratulations!